# Cryolipolysis-induced abdominal fat change: Split-body trials

**In Cheol Hwang, Kyoung Kon Kim[‡], Kyu Rae Lee** [ID] *

Family Medicine, Gachon University, Incheon, Korea

‡ Co-authored as a first author.
* baria07@daum.net

**Data Availability Statement:** All relevant data are within its Supporting Information files.

**Funding:** This study was fully supported by a grant from the Gachon University research fund of 2018. (GCU-2018-5258). The funding agency had no role in study design, data collection and analysis,

## Abstract

Cryolipolysis has been considered as a noninvasive alternative to surgical procedures for reducing subcutaneous fat without affecting the surrounding tissues. However, no clinical trial has investigated changes in the abdominal fat tissue by 12 weeks after cryolipolysis. Therefore, in this split-body trial, we explored whether a single session of unilateral cryolipolysis could change visceral and subcutaneous adipose tissue over a period of 12 weeks. We compared the cross-sectional areas of the abdominal adipose tissue of 15 subjects (9 women; 38.3 [10.8] years) by computed tomography before and at 12 weeks after a single treatment of cryolipolysis to the left abdomen and used the right abdomen as untreated control. In addition, we measured participants' waist circumference, percentage of body fat (by bioelectrical impedance analysis) at baseline and at 6- and 12-weeks post-treatment. Single unilateral cryolipolysis tended to reduce the cross-sectional areas of visceral adipose tissue, by 8.4 cm$^2$ (9.9%), the waist circumferences, and the percent body fat, by 2.8 cm$^2$ (0.6%), overall. The cross-sectional area of visceral adipose tissues on the treated side significantly decreased, by 6.8 cm$^2$ (15.6%; P = 0.003), and that of the untreated side tended to decrease by 1.2 cm$^2$ (3.6%). Thus, a single unilateral session of noninvasive selective cryolipolysis can be considered as a safe and effective treatment for reduction of visceral adipose tissue over a period of 12 weeks, which should result in metabolic improvement.

## Introduction

Visceral obesity has been recognized as a major culprit to metabolic disorders such as insulin resistance, type 2 diabetes mellitus, and metabolic syndrome. Visceral adipose tissue (VAT) is considered as a major risk fat depot [1]. VAT, subcutaneous adipose tissue (SAT) vary in quality are considered even greater cardiometabolic risk variables than body mass index (BMI) [2, 3].

Previous studies have shown that liposuction does not improve coronary metabolic risk in individuals with abdominal obesity [4]. Surgical approaches such as liposuction and lipectomy focus on removal of SAT, rather than VAT [5] and pose risks of infection, bleeding, and postoperative scarring [6]. In addition, operative procedures have some disadvantages, such as high cost, longer recovery, and postoperative complications, and even mortalities, without providing metabolic benefit [4].

decision to publish, or preparation of the manuscript.

**Competing interests:** No authors have competing interests

Since FDA approval in 2010, cryolipolysis has been considered as a noninvasive alternative to surgical procedure for reducing subcutaneous fat without affecting the surrounding tissues [7–9].

This procedure causes apoptotic fat cell death and reduces the thickness of the subcutaneous fat. When used for the reduction of subcutaneous adnominal fat, it is not associated with changes in serum lipids or liver test results [10, 11].

There is evidence that adipose tissue is selectively sensitive to cold injury, for instance, "popsicle panniculitis," where cytoplasmic lipids in adipocytes crystalize at temperatures well above the freezing point of tissue water. Manstein et al. first reported that black Yucatan pigs showed grossly visible loss of 3.5-mm thickness of subcutaneous fat after cold-induced tissue injury [12]. A single cycle of cryolipolysis to cold resulted in macrophage recruitment over a period of 2 weeks, which peaked at 4 weeks, and disappeared by about 12 weeks after treatment. Panniculitis may further augment the damage to adipocytes in the early period and subsequent loss of fat [13].

Selective cryolipolysis has been introduced as an effective method for treating local fat and SAT in human clinical trials to date. The clinical efficacy of cryolipolysis has been evaluated by comparing photographs or by measuring superficial fat thickness with sonography after treatment, but its role in the VAT and SAT fields has not been elucidated to date.

Therefore, we here evaluated the clinical efficacy and safety of cryolipolysis for reducing body fat, and waist and abdominal adipose tissue (VAT and SAT) after unilateral cryolipolysis treatment of the left abdomen in a split-body trial method in healthy Koreans over a 12-week period.

## Material and methods

The institutional review board of Gachon University approved the study protocol (GCIRB2018-320, September 11, 2018) and all participants provided written consent. This study was registered at cris.nih.go.kr (number KCT0003647).

### Subjects

We used G-power software to determine the sample size to obtain an effector size of 1 with power (1-beta error probability), 2-tailed, $\alpha = 0.05$, non-centrality parameter $\delta = 3.9088201$, critical $t = 2.1408635$, degree of freedom = 14.2788745. A total sample size of 16 was required [14]. Consequently, we enrolled 19 subjects through indoor bulletins and street flyers between September 12 and November 20, 2018 and informed them of the study protocol before the trial.

We included healthy men or women without clinical conditions, with BMI $\geq 20$ kg/m$^2$, and those without a change of more than 5% of their weight over the previous 6 months. Those who had a history of cold-related diseases, such as cryoglobulinemia and cold hemogobulinemia, metabolic abnormalities, such as diabetes, dyslipidemia, or thyroid disease, or had used any medication to regulate weight over the previous 6 months were excluded. In addition, we did not include pregnant or lactating women.

As it is not possible select cases matched in terms of body composition (VAT, SAT, body fat), we performed sequential split-body trials were performed to compare the effect of cryolipolysis by examining cross-sectional areas of adipose tissues. We assessed the height and weight of participants using a digital portable standiometer (DS-102, Jenix, Seoul, Korea), and determined their percentage of body fat by bioelectrical impedance analysis (Biospace, Seoul, Korea). Waist circumference was measured at the anterior superior iliac spine level at baseline,

and at 6 and 12 weeks after treatment [15]. All procedures were performed between November 27, 2018, and February 28, 2019, after overnight fasting.

The template for the CONSORT flow diagram is shown in Fig 1.

### Cryolipolysis

A new cryolipolysis machine (Cryo-Elsa, Huons Co., Ltd., Seongnam, Korea) was approved by the Korean FDA in 2017. The device consists of a $473 \times 756 \times 1200$-mm-sized control unit (main body), gel pack, and two vacuum applicators, which are lined to maintain a low temperature of the skin, and alternatively cool and massage the skin in the targeted area. It applies evenly controlled chilling of the adipose tissue through the skin. The applicator head uses a mild vacuum to retain the tissue between two cooling plates within the cup for 1 hour to destroy the fat cells.

Precisely controlled cooling was applied to the left abdomen and the corresponding area in the back, with one vacuum applicator placed at the umbilicus level, by the same operator in all participants. The right abdomen was not treated, as control. We performed a single treatment at the left abdomen, at –7˚C for 60 minutes, as previously reported [9].

### Computed tomography

We compared the cross-sectional areas of VAT and SAT between the treatment (left) and control (right) abdomen by computed tomography (CT) at the first lumbar vertebra level, both at pretreatment baseline and at 12 weeks after treatment. We used fat Pointer software to calculate the area of tissue with a CT value corresponding to visceral fat (red color) and subcutaneous fat (blue color) regions on abdominal CT images (Spuria, Hitachi, Tokyo, Japan).

### Lifestyle evaluations

We instructed all participants to maintain their behavioral lifestyle, such as meal patterns and activity levels, as usual during the trial. They submitted 4-day food diaries to the study nutritionist at the 6th and 12th week visits. We calculated their food intake from these diaries using Can-pro software (Korea Nutrition Society).

### Adverse events

We monitored the safety of the treatment through documentation of adverse events and clinical assessment of the treatment sites for symptoms such as numbness [16], pain, erythema, abnormal hyperplasia, and any persistent dermatological findings by interview at the 6- and 12-week visits.

**Statistical analysis.**   We performed the Wilcoxon signed-rank test (matched pairs) to assess differences in waist circumference, body fat percentage, the areas of VAT and SAT, and the visceral-to-subcutaneous fat ratio (VS ratio) between baseline and 12 weeks. SPSS for Windows (version 18; SPSS Inc., Chicago, IL, USA) was used for statistical analysis. Two tailed p-values $< 0.05$ were considered to indicate statistical significance.

## Results

### Basic characteristics

Of the 19 subjects, 6 were women. Two subjects withdrew from the study and 2 subjects were lost to follow-up; thus, 15 subjects completed the study (mean age: $38.31 \pm 10.84$ years; mean BMI: $25.64 \pm 2.76$ kg/m$^2$) completed the study. With these 15 subjects, the power (1 - β error probability) was calculated to be exceed 0.93.

## CONSORT Flow Diagram

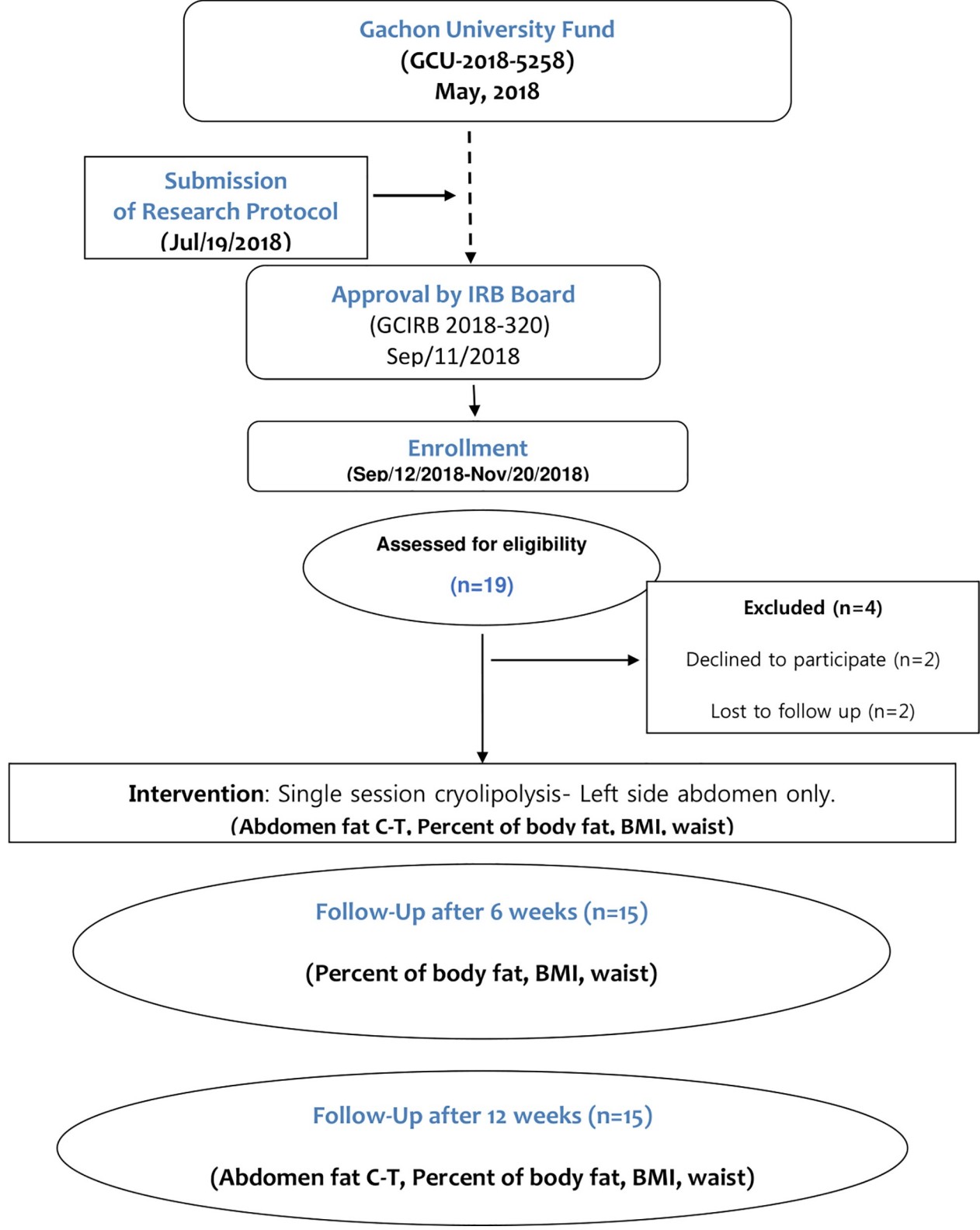

**Fig 1. Participant flow chart.**

**Table 1. Change of anthropometric, waist, and cross-sectional areas of adipose tissue.**

|  | Initial visit | 6-week visit | 12-week visit | P-value (Initial–12-week) |
|---|---|---|---|---|
| Weight (kg) | 69.88 (9.37) | 70.52 (9.27) | 69.49 (9.34) | 0.637 |
| BMI (kg/m$^2$) | 25.64 (2.76) | 25.87 (2.75) | 25.47 (2.53) | 0.700 |
| Waist Circumference (cm) | 91.18 (7.87) | 90.80 (7.81) | 88.52 (10.66) | 0.294 |
| Waist-to-Hip Ratio | 0.91 (0.04) | 0.92 (0.05) | 0.91 (0.05) | 0.859 |
| Percentage of body fat (%) | 33.43 (5.83) | 33.29 (5.92) | 32.79 (5.65) | 0.187 |
| Cross-sectional Areas of Visceral Fat (cm$^2$) | 84.31 (33.60) | - | 75.94 (34.65) | 0.057 |
| Cross-sectional Areas of Subcutaneous Fat (cm$^2$) | 216.35 (68.79) |  | 217.95 (69.01) | 0.820 |
| Visceral-to-Subcutaneous Fat Ratio | 0.42 (0.19) | - | 0.37 (0.18) | 0.078 |
| Visceral-to-Total Fat Ratio | 0.29 (0.09) | - | 0.26 (0.09) | 0.061 |

The characteristics of the 15 subjects are shown in Table 1. There were no significant statistical differences in weight (p = 0.637) or BMI (p = 0.700) between the baseline and 12-week visits. There were also no statistically significant differences in the percentage body fat (p = 0.609) or the waist circumference (p = 0.294) between the baseline and 12-week visits (Table 1).

## Effect of cryolipolysis

The cross-sectional area of the VAT overall showed a tendency to decrease, by 9.9% (p = 0.057), while that of the SAT overall increased non-significantly, by 0.7% (p = 0.820) over the 12-weeks period after a single cryolipolysis session (Figs 2 and 3).

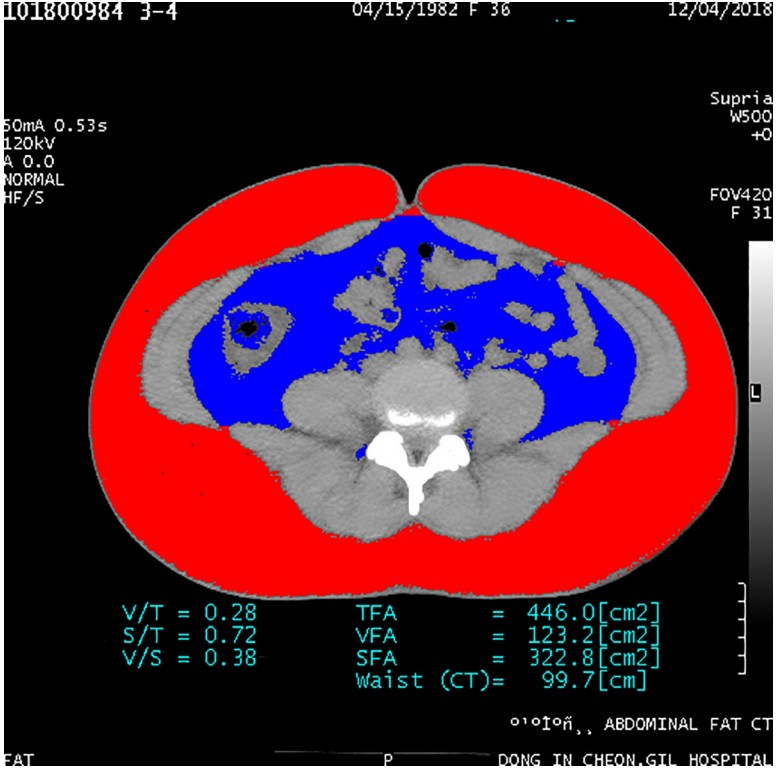

**Fig 2. The cross-sectional areas of overall visceral and subcutaneous adipose tissues at the initial visit.**

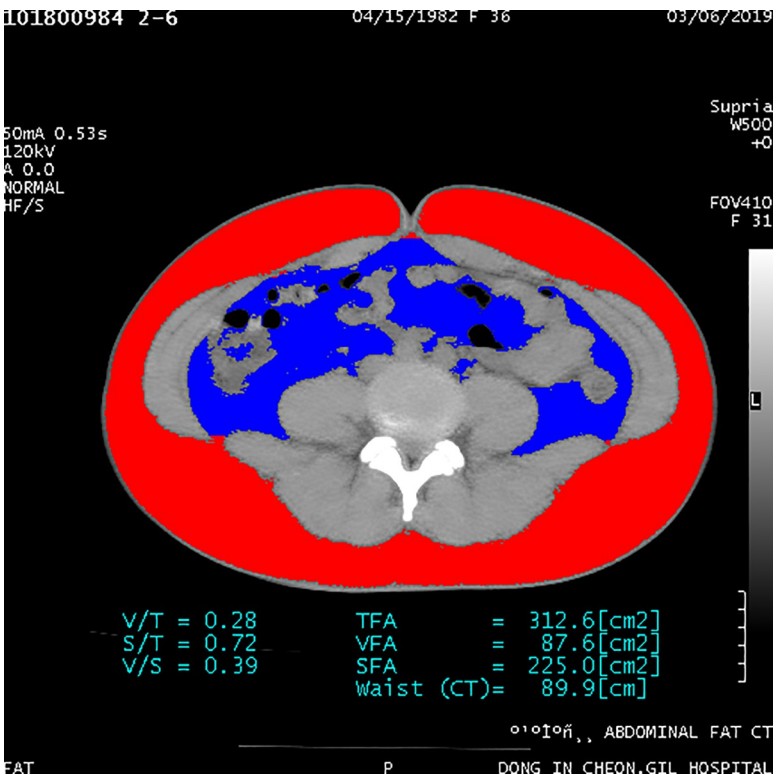

**Fig 3. The cross-sectional areas of overall visceral and subcutaneous adipose tissues at the 12-week visit.**

The cross-sectional area of VAT in the treated left abdomen decreased highly significantly, by 15.6% (6.8 cm$^2$), from 43.53 ± 17.55 cm$^2$ to 36.67 ± 18.08 cm$^2$ (p = 0.003), while that of the untreated right abdomen decrease non-significantly, from 40.74 ±16.47 cm$^2$ to 39.18 ± 17.02 cm$^2$ (p = 0.281). The cross-sectional area of SAT in the treated left abdomen changed non-significantly from 107.49 ± 35.54 cm$^2$ to 109.04 ± 34.44 cm$^2$ (p = 0.733), and that of the control right abdomen changed non-significantly from 108.47. ± 33.85 cm$^2$ to 108.93. ± 34.83 cm$^2$ (p = 0.865) (Fig 4).

Moreover, the single-session cryolipolysis tended to decrease the VS ratio from 0.419 ± 0.19 to 0.37 ± 0.18 in (p = 0.078), and the visceral-to-total fat ratio from 0.28 ± 0.09 to 0.26 ± 0.09 (p = 0.061) over the 12-week period (Fig 5).

According to daily food intake, as assessed from food diaries, there was no statistical difference in the mean calculated calories at the initial, 6-week, and 12-week visits (1300–1576 kcal/day). Transient minor adverse events, such as pain, numbness, and transient bruising, were noted within a week, but no permanent consequences were noted.

## Discussion

As no clinical trial to date has investigated the change in abdominal fat tissue after cryolipolysis, we here explored if a single-session of unilateral cryolipolysis could change VAT and SAT over a period of 12 weeks, in a split-body trial. We showed that the cross-sectional area of the VAT in the treatment side of the abdomen decreased significantly, by 6.8 cm$^2$ (15.6%; P = 0.003), while that of the untreated side showed a decrease of 1.2 cm$^2$ (3.6%). Moreover, no paradoxical adipose hyperplasia, which has been reported previously [17], was noted, indicating the safety of the procedure.

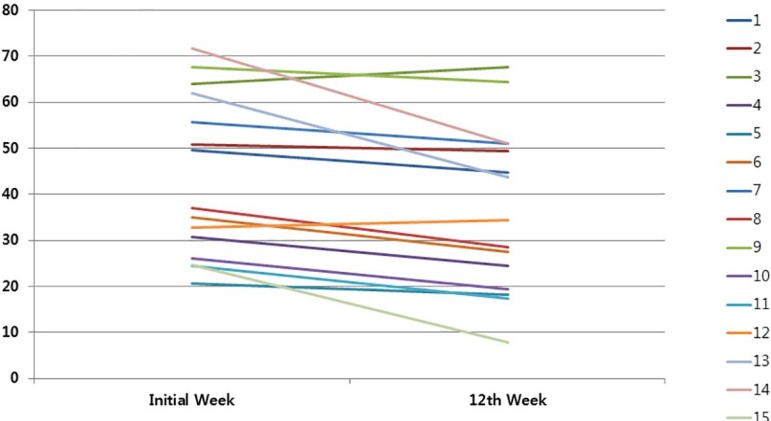

**Fig 4. Change in cross-sectional areas of left visceral adipose tissues.**

VAT is a hormonally active element of body fat, which possesses distinctive biochemical characteristics related to metabolically pathological processes in humans. Excessive VAT has been associated with coronary artery disease, insulin resistance, diabetes, and hypertension [18]. The quantitative measurement of VAT is pivotal for evaluating the potential risk for the development of these pathologies, as well as to inform accurate prognosis [19]. In addition, a previous clinical trial had shown reduction of superficial fat through sonographic devices, photographic appraisal, and caliper measurement [20]. A previous descriptive study of 170 Asian cases reported a 5.3 (1.7) mm (23.2%) reduction in fat thickness by high-resolution ultrasonography after cryolipolysis to the abdomen [21]. In addition, sonographic evaluation of intra-abdominal adipose tissue yields a coefficient of variation of 64%; therefore, a previous paper did not recommend ultrasound for the assessment of VAT [22]. Thus, CT is considered the

# Change of V/S, V/T fat Ratio

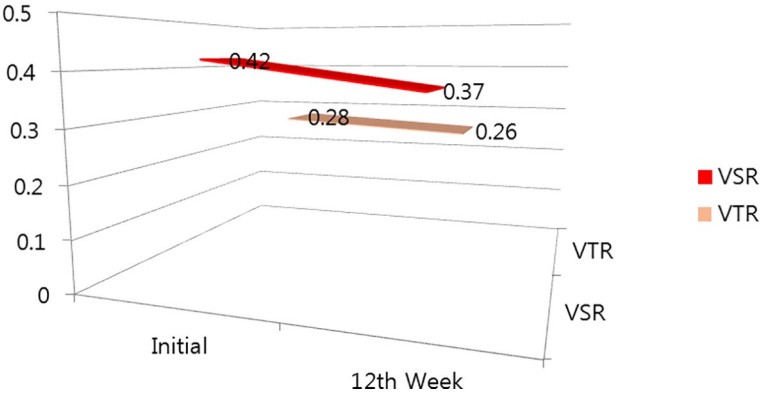

**Fig 5. Comparison of the visceral-to-subcutaneous fat ratio and visceral-to-total fat ratio.**

gold standard for quantitative measurement of the cross-sectional areas of intra-abdominal adipose tissue.

A previous study reported that a single session of both lateral cryolipolysis to the abdomen over both "love handles" showed a significant reduction in VAT, by 16.2 cm$^2$ (11.4%), over a period of 8 weeks [23]. This was comparable to our findings of a 15.64% reduction VAT cross-sectional area on the treated abdomen compared to that in the control abdomen (1.2 cm$^2$, 3.6%). The markers for central fat, such as percentage of body fat and waist circumference, tended to decrease because the single treatment significantly reduced the cross-sectional areas of VAT on the treated side of the abdomen. Nevertheless, the clinical effect of cryolipolysis was less effective than that of liposuction in terms of total fat reduction. Although the difference did not reach statistical significance, the cryolipolysis used here might still be effective in reducing central adiposity (waist circumference) and percent of body fat.

The visceral-to-subcutaneous adipose tissue ratio (VS ratio) is correlated with cardio-metabolic risk, more so than BMI. The susceptibility to store fat viscerally versus subcutaneously may be a distinctive risk factor, independent of the absolute fat amount [24]. The VS ratio has been considered to be one of the markers of metabolic derangement in cardiovascular diseases. Liposuction has led to metabolic worsening, such as an increase in the VS ratio (0.50 [1713/3414] to 0.89 [1673/1895]) in the normal glucose tolerance group, even in the type 2 diabetes mellitus group (0.69 [2653/3803] to 0.88 [2425/2751]). Because liposuction decreases subcutaneous adipose tissues by 28–44%, large-volume abdominal liposuction did not demonstrate significant improvement in obesity-related metabolic risk variables [25]. In contrast, single unilateral treatment by cryolipolysis significantly decreased the visceral fat of the treated abdomen and thereby tended to decrease the VS ratio (0.42 [84.3/216.4] to 0.37 [75.9/217.9]) in apparently healthy normal subjects in this study. Thus, the single unilateral session of cryolipolysis tended to improve the VS ratio over large-volume liposuction over a 12-week period. A recent clinical trial was performed to represent visual imaging outcomes related to metabolic changes in SAT through direct optical spectroscopic imaging after cryolipolysis [26].

A recent study in swines demonstrated that cold (-4.8°C; 2-cm depth) icy slurry injection induced direct damage to adipocytes by lipid crystallization [27]. Another animal experiment showed that cold-induced tissue remodeling is more evident in VAT than SAT, which is characterized by a scarcity of inflammatory macrophages, implicating inflammatory cells in the process [28]. Visceral adiposity correlated strongly with insulin resistance and metabolic derangement. The previous experiment showed induction of adipocyte macrophage M2 and adipogenic progenitors, which alleviate meta-inflammation, in the VAT, by cold. In addition, another in vivo study demonstrated strong induction of the mRNA of the VAT-derived serine protease inhibitor (vaspin) mRNA in brown adipose tissue of cold-exposed mice [29]. Therefore, cooling may reduce deep adipose tissue, such as VAT, and cold-induced sympathetic stimulation of VAT in recruited adipocyte M2 macrophages and alleviated chronic meta-inflammation, while inducing tissue remodeling and rare adipose tissue browning in mice. In addition, another clinical human cadaver study in Siberia showed a higher percentage of brown-like adipocytes, with more intense UCP 1 immuno-reactive cells in visceral fat [30]. However, it remains to be clarified whether cold exposure of VAT would result in the same metabolic remodeling in living humans; hence, large-scale controlled human clinical trials should be considered in future.

This study had several limitations. Histologic studies would be better to prove chemical reactions, such as [popsicle panniculitis, in the 12-week period. Second, a larger clinical trial, using parametric paired $t$-tests, would be necessary to produce more powerful and reliable statistics. Third, uniform demographic participants, such as individuals of the same sex or age would help to determine the effectiveness of the cooling method in the sub-populations.

Fourth, we did not measure metabolic chemical profiles, such as adipocytokines, interleukin, and vaspin, which are metabolic risk factors.

## Conclusions

In conclusion, a single, unilateral, noninvasive, selective cryolipolysis session could be considered as a safe and effective treatment for the reduction of VAT with metabolic improvement.

## Supporting information

**S1 Checklist. TREND statement checklist.**
(PDF)

**S1 File. Research proposal.**
(PDF)

**S2 File.**
(XLSX)

## Author Contributions

**Conceptualization:** Kyu Rae Lee.

**Data curation:** Kyoung Kon Kim, Kyu Rae Lee.

**Formal analysis:** Kyu Rae Lee.

**Funding acquisition:** Kyu Rae Lee.

**Investigation:** Kyu Rae Lee.

**Methodology:** Kyu Rae Lee.

**Project administration:** Kyu Rae Lee.

**Resources:** Kyu Rae Lee.

**Software:** Kyu Rae Lee.

**Supervision:** In Cheol Hwang, Kyu Rae Lee.

**Validation:** Kyu Rae Lee.

**Visualization:** Kyu Rae Lee.

**Writing – original draft:** Kyu Rae Lee.

**Writing – review & editing:** In Cheol Hwang, Kyu Rae Lee.

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
