## [Decision Letter · Decision Letter 0]

31 Jul 2020

PONE-D-20-06167

Variation of Percent of Body fat, Waist, and Cross Sectional Areas of Abdominal Adipose Tissue after Single Unilateral Cryolipolysis-Using the sequential split-body trials-

PLOS ONE

Dear Dr. lee,

Thank you for submitting your manuscript to PLOS ONE. After careful consideration, we feel that it has merit but does not fully meet PLOS ONE’s publication criteria as it currently stands. Therefore, we invite you to submit a revised version of the manuscript that addresses the points raised during the review process.

The reviewers have  raised several area that need to be addressed including both methodological and statistical issues. The manuscript needs to be clearly written and particularly with a more focused discussion, and be proof read by a native English speaker.

We look forward to receiving your revised manuscript.

Kind regards,

Stephen L Atkin, MD

Academic Editor

PLOS ONE

Journal Requirements:

2. Please include in your Methods section the date ranges over which you recruited participants to this study.

3. Thank you for submitting your clinical trial to PLOS ONE and for providing the name of the registry and the registration number. The information in the registry entry suggests that your trial was registered after patient recruitment began. PLOS ONE strongly encourages authors to register all trials before recruiting the first participant in a study.

1) your reasons for your delay in registering this study (after enrolment of participants started);

2) confirmation that all related trials are registered by stating: “The authors confirm that all ongoing and related trials for this drug/intervention are registered”.

Please also ensure you report the date at which the ethics committee approved the study as well as the complete date range for patient recruitment and follow-up in the Methods section of your manuscript.

"This study was supported by the Gachon University research fund of 2018. (GCU-2018-5258)"

"Yes, the funder's specification was described in the main text file."

6. Please amend your manuscript to include your abstract after the title page.

7. Please include a separate caption for each figure in your manuscript.

8. Please include your tables as part of your main manuscript and remove the individual files. Please note that supplementary tables (should remain/ be uploaded) as separate "supporting information" files.

Reviewers' comments:

Reviewer's Responses to Questions

**Comments to the Author**

1. Is the manuscript technically sound, and do the data support the conclusions?

Reviewer #1: Yes

Reviewer #2: Partly

Reviewer #3: Partly

2. Has the statistical analysis been performed appropriately and rigorously? 

Reviewer #1: Yes

Reviewer #2: No

Reviewer #3: Yes

3. Have the authors made all data underlying the findings in their manuscript fully available?

Reviewer #1: Yes

Reviewer #2: Yes

Reviewer #3: Yes

4. Is the manuscript presented in an intelligible fashion and written in standard English?

Reviewer #1: Yes

Reviewer #2: Yes

Reviewer #3: Yes

5. Review Comments to the Author

Reviewer #1: 1. There are grammatical errors throughout. Please have native English speaker review and edit, and correct syntax, diction, tense, and punctuation. For example, in intro you say that VATS and SATS are "various." This is not correct english. There are more errors throughout the paper. Not a "weakness" but a risk, or disadvantage, etc.

2. The title needs to be improved. It's not the variation you are presenting, but the effect of cryolipolysis on adipose tissue. There's a difference in the meaning of the word "variation."

3. BMI should be spelled out in its first appearance in the manuscript.

4. Your paper has many strengths, but in its current form with sentence structure and english writing errors it makes it difficult to read.

5. Why would visceral fat change if the cryolipolysis is only on the subcutaneous fat?

6. How do you determine metabolism of the fat?

7. Does the patients medical condition, such as Diabetes or heart condition improve as a result of decreased VATS?

8. How do you control for anatomic variability with visceral fat, since the internal anatomy of a patient may change between CT scan sessions?

Overall, the paper has merit and several strengths, but needs more work to present the information in a more coherent fashion.

Reviewer #2: The manuscript entitled ‘Variation of Percent of Body fat, Waist, and Cross Sectional Areas of Abdominal Adipose Tissue after Single Unilateral Cryolipolysis-Using the sequential split-body trials’ with the aim to explore if single session unilateral cryolipolysis on left abdomen could change visceral and subcutaneous adipose tissue (VAT/SAT) in 12 weeks.

The manuscript can be further improved based on the comments below.

Abstract

Mean, sd to be stated for the figures. Figures for men to be included. P value for 1.2 cm2 (3.6%) to be stated. cm2 to be written as cm^2 (2 in superscript form).

Materials and Methods

Page 4, title 'Material and Method' to be stated as Materials and Methods.

Page 4, the statement ‘All participants were provided with written informed consents.’ requires revision.

Page 4, sample size calculation write up requires revision. More input parameters to be provided i.e 1 or 2 tailed, alpha (0.05), power of study etc. Non-centrality parameter δ 4, critical t 2.314495, degree of freedom 15, total sample size 16 are output parameters.

Page 5, statistical analysis section to be placed at the end of the materials and methods section. Paired t test for what comparison to be clearly stated e.g. Initial week and 12-week.

Results

Page 6, the word mean, sd to be stated where applicable.

Page 7, p value to be stated for all results/findings.

Table 1, the name of the statistical test to be denoted in the table footnote as well as in the statistical analysis section. For the Waist to Hip Ratio, was the value for the SD (initial visit and 8 weeks) zero or there was a value at 2nd decimal point onward? .9 to be stated as 0.9 Decimal point for the p values to be standardized. The dark background to be discarded and replaced with words.

Presentation format of .0 or 0. for figures to be standardized throughout the manuscript [text, table, figure(s)].

Active sentences to be written in passive form.

List of references and citation of references in the text did not meet the journal format.

Reviewer #3: The study presents an orderly and very clear methodology, with interesting results. But unfortunately, the discussion does not clearly present the effect of cryolipolysis on visceral fat, nor does it justify this effect, making the necessary correlation with the literature.

6. PLOS authors have the option to publish the peer review history of their article (what does this mean?). If published, this will include your full peer review and any attached files.

Reviewer #1: No

Reviewer #2: No

Reviewer #3: No

---

## [Author Response · Author response to Decision Letter 0]

27 Aug 2020

1. There are grammatical errors throughout. Please have native English speaker review and edit, and correct syntax, diction, tense, and punctuation. For example, in intro you say that VATS and SATS are "various." This is not correct english. There are more errors throughout the paper. Not a "weakness" but a risk, or disadvantage, etc.

I have had the manuscript checked by a native English speaker. 

2. The title needs to be improved. It's not the variation you are presenting, but the effect of cryolipolysis on adipose tissue. There's a difference in the meaning of the word "variation."

I have modified the title according to your recommendation.

3. BMI should be spelled out in its first appearance in the manuscript.

I have spelled out body mass index (BMI) on its first appearance following your advice.

4. Your paper has many strengths, but in its current form with sentence structure and english writing errors it makes it difficult to read.

I have had the manuscript checked by a native English speaker. 

5. Why would visceral fat change if the cryolipolysis is only on the subcutaneous fat?

Visceral adiposity is known for being a metabolic risk depot related to T2DM and dyslipidemia, according to recent animal studies. 

Our hypothesis is that cold induced browning of visceral fat is possible, similar to that of subcutaneous fat. 

6. How do you determine metabolism of the fat?

In general, most published studies have measured adipocytokines, such as interleukin and caspase from the fat. However, we did not check the chemical profile, but determined the expressed visceral to subcutaneous ratio and used this as a metabolic parameter. This has been described in the last part of the Discussion section. 

7. Does the patients medical condition, such as Diabetes or heart condition improve as a result of decreased VATS?

All the subjects were recruited without any metabolic abnormalities, such as DM or thyroid diseases. This has been mentioned in the Methods and Materials section.

8. How do you control for anatomic variability with visceral fat, since the internal anatomy of a patient may change between CT scan sessions?

We measured the cross-sectional areas of adipose tissues at the umbilicus level (first lumbar vertebra) after fasting for 8 hours, before beginning the study and at 12 weeks.

Overall, the paper has merit and several strengths, but needs more work to present the information in a more coherent fashion.

Reviewer #2: The manuscript entitled ‘Variation of Percent of Body fat, Waist, and Cross Sectional Areas of Abdominal Adipose Tissue after Single Unilateral Cryolipolysis-Using the sequential split-body trials’ with the aim to explore if single session unilateral cryolipolysis on left abdomen could change visceral and subcutaneous adipose tissue (VAT/SAT) in 12 weeks.

The manuscript can be further improved based on the comments below.

Abstract

Mean, sd to be stated for the figures. Figures for men to be included. P value for 1.2 cm2 (3.6%) to be stated. cm2 to be written as cm^2 (2 in superscript form).

Materials and Methods

Page 4, title 'Material and Method' to be stated as Materials and Methods.

Page 4, the statement ‘All participants were provided with written informed consents.’ requires revision.

Page 4, sample size calculation write up requires revision. More input parameters to be provided i.e 1 or 2 tailed, alpha (0.05), power of study etc. Non-centrality parameter δ 4, critical t 2.314495, degree of freedom 15, total sample size 16 are output parameters.

Page 5, statistical analysis section to be placed at the end of the materials and methods section. Paired t test for what comparison to be clearly stated e.g. Initial week and 12-week.

Results

Page 6, the word mean, sd to be stated where applicable.

Page 7, p value to be stated for all results/findings.

Table 1, the name of the statistical test to be denoted in the table footnote as well as in the statistical analysis section. For the Waist to Hip Ratio, was the value for the SD (initial visit and 8 weeks) zero or there was a value at 2nd decimal point onward? .9 to be stated as 0.9 Decimal point for the p values to be standardized. The dark background to be discarded and replaced with words.

Presentation format of .0 or 0. for figures to be standardized throughout the manuscript [text, table, figure(s)].

Active sentences to be written in passive form.

List of references and citation of references in the text did not meet the journal format.

Thank you for your review and the detailed comments. All the corrections have been made per your recommendations.

Reviewer #3: The study presents an orderly and very clear methodology, with interesting results. But unfortunately, the discussion does not clearly present the effect of cryolipolysis on visceral fat, nor does it justify this effect, making the necessary correlation with the literature.

Thank you for your comments.

I will revise the manuscript and highlight the effects of cryolipolysis on visceral fat. I also aim to support my hypothesis with the required references.

---

## [Decision Letter · Decision Letter 1]

10 Nov 2020

Cryolipolysis induced abdominal fat change -split-body trials-

PONE-D-20-06167R1

Dear Dr. Lee,

We’re pleased to inform you that your manuscript has been judged scientifically suitable for publication and will be formally accepted for publication once it meets all outstanding technical requirements.

Kind regards,

Stephen L Atkin, MD

Academic Editor

PLOS ONE

Additional Editor Comments (optional):

Reviewers' comments:

Reviewer's Responses to Questions

**Comments to the Author**

1. If the authors have adequately addressed your comments raised in a previous round of review and you feel that this manuscript is now acceptable for publication, you may indicate that here to bypass the “Comments to the Author” section, enter your conflict of interest statement in the “Confidential to Editor” section, and submit your "Accept" recommendation.

Reviewer #2: All comments have been addressed

Reviewer #3: All comments have been addressed

2. Is the manuscript technically sound, and do the data support the conclusions?

Reviewer #2: Partly

Reviewer #3: Yes

3. Has the statistical analysis been performed appropriately and rigorously? 

Reviewer #2: Yes

Reviewer #3: N/A

4. Have the authors made all data underlying the findings in their manuscript fully available?

Reviewer #2: Yes

Reviewer #3: Yes

5. Is the manuscript presented in an intelligible fashion and written in standard English?

Reviewer #2: Yes

Reviewer #3: Yes

6. Review Comments to the Author

Reviewer #2: Table 1 & Table 1 Rev

Visceral-to-Subcutaneous Fat Ratio - add 0 in front of dot

Visceral-to-Total Fat Ratio - add 0 in front of dot

Reviewer #3: All the doubts from the reviwers were response by the authors. I believe that the article is prepared for publication.

7. PLOS authors have the option to publish the peer review history of their article (what does this mean?). If published, this will include your full peer review and any attached files.

Reviewer #2: No

Reviewer #3: No

---

## [Editor Report · Acceptance letter]

1 Dec 2020

PONE-D-20-06167R1 

Cryolipolysis-induced abdominal fat change
-split-body trials- 

Dear Dr. Lee:

I'm pleased to inform you that your manuscript has been deemed suitable for publication in PLOS ONE. Congratulations! Your manuscript is now with our production department. 

Kind regards, 

on behalf of

Dr. Stephen L Atkin 

Academic Editor

PLOS ONE